# Nanostructured Calcium-Incorporated Surface Compared to Machined and SLA Dental Implants—A Split-Mouth Randomized Case/Double-Control Histological Human Study

**DOI:** 10.3390/nano13020357

**Published:** 2023-01-16

**Authors:** Christian Makary, Abdallah Menhall, Pierre Lahoud, Hyun-Wook An, Kwang-Bum Park, Tonino Traini

**Affiliations:** 1Oral Surgery Department, Saint Joseph University, Beirut P.O. Box 1104-2020, Lebanon; 2Department of Dental Science, Kyungpook National University, Daehak-ro, Buk-gu, Daegu 41566, Republic of Korea; 3Daegu Mir Dental Hospital, Jung-gu, Daegu 41934, Republic of Korea; 4Department of Innovative Technologies in Medicine & Dentistry, University “G. d’Annunzio” of Chieti-Pescara, 66100 Chieti, Italy; 5Electron Microscopy Laboratory, University “G. d’Annunzio” of Chieti-Pescara, 66100 Chieti, Italy

**Keywords:** nano surfaces, early bone formation, implant surface, histomorphometry, osseointegration

## Abstract

Background: Implant surface topography is a key element in achieving osseointegration. Nanostructured surfaces have shown promising results in accelerating and improving bone healing around dental implants. The main objective of the present clinical and histological study is to compare, at 4 and 6 weeks, (w) bone-to-implant contact in implants having either machined surface (MAC), sandblasted, large grit, acid-etched implant surface (SLA) medium roughness surface or a nanostructured calcium-incorporated surface (XPEED^®^). Methods: 35 mini-implants of 3.5 × 8.5 mm with three different surface treatments (XPEED^®^ (*n* = 16)—SLA (*n* = 13)—MAC (*n* = 6), were placed in the posterior maxilla of 11 patients (6 females and 5 males) then, retrieved at either 4 or 6w in a randomized split-mouth study design. Results: The BIC rates measured at 4w and 6w respectively, were: 16.8% (±5.0) and 29.0% (±3.1) for MAC surface; 18.5% (±2.3) and 33.7% (±3.3) for SLA surface; 22.4% (±1.3) and 38.6% (±3.2) for XPEED^®^ surface. In all types of investigated surfaces, the time factor appeared to significantly increase the bone to implant contact (BIC) rate (*p* < 0.05). XPEED^®^ surface showed significantly higher BIC values when compared to both SLA and MAC values at 4w (*p* < 0.05). Also, at 6w, both roughened surfaces (SLA and XPEED^®^) showed significantly higher values (*p* < 0.05) than turned surface (MAC). Conclusions: Nanostructured Calcium titanate coating is able to enhance bone deposition around implants at early healing stages.

## 1. Introduction

The first titanium dental implant placed by Brånemark in a human volunteer in 1965 had a cylindrical shape and a machined surface [1]. Since that time, the main focus of clinicians and researchers has been to obtain a better quality of osseointegration, the functional bone formation in direct contact with the implant [2]. Owing to these improvements, the modern-day titanium implant offers much higher success rates and substantially faster function than its original predecessor [3]. After implantation, direct bone apposition onto the fixture is critical for the successful loading of dental implants [4]. Surface treatment plays a major role in this biological integration [5]. Since the early phases of implantology, important alterations in the commercially available implants have been made, many of which involved different methods of surface roughening [6]. However, the “optimal” surface type is yet to be determined, quality and speed of osseointegration being the main criteria [7].To improve the favorable biologic response to titanium and decrease healing time for osseointegration, different implant surface modifications have been introduced, treatment methods have been applied to improve the surface topography and properties, such as sandblasting, acid etching, anodization, discrete calcium-phosphate crystal deposition, coatings with biologic molecules, and chemical modification [7,8,9]. Also, natural polymer coatings are being increasingly experimented upon in that field, for their availability, versatility, biodegradability and biocompatibility [10]. These surface topography modifications have yielded significant results and the great majority of published articles have reported increased bone fixation and increased bone-to-implant contact for rougher implant surfaces compared to polished, milled, or turned surfaces [9,10,11,12]. Mechanical treatments of the implant surface, such as media blasting (BM) and titanium plasma spraying (TPS), have been shown to increase roughness, therefore yielding higher contact surface with surrounding bone [13]. However, these techniques have shown limitations due to transmission of residue particles, such as TPS coating and non-resorbable blast media (TiO_2_ or Al_2_O_3_) into surrounding bone and tissues, which might interfere with the osseointegration process [14]. Also, this type of surface treatment was shown to increase the risk of infection and peri-implantitis [15]. Medium-roughness surfaces were introduced to enhance bone healing around implants while reducing bacterial contamination and peri-implantitis [16]. Therefore, sandblasting followed by chemical surface treatment such as acid-etching was another method introduced for roughening titanium dental implants; it produces micro pits on titanium surfaces and has been shown to greatly enhance osseointegration [8,14,17,18]. Sandblasting with large grit corundum followed by acid etching with sulfuric and hydrochloric acid surfaces (SLA) were introduced and seemed to yield promising results with higher success rates and faster loading time [18,19,20]. Also, a recent study showed that this type of surface responded better to chemical-mechanical treatment methods when peri-implantitis occurs [21]. However, while implant microtopography seemed to affect osseointegration at the cellular level, nanotopography was thought to influence cell-implant interactions at both cellular and protein level, making nano-scale implant surface characteristics an increasingly growing area of interest for biomedical engineers [22,23,24]. More recently, and in order to improve osseointegration and reduce healing time, Titanium implants with a nanostructured calcium-incorporated surface have been introduced [23,25]. In a systematic review of implant coatings, it was determined that the proper materials and preparation techniques are quintessential in implant success and play a major role in enhancing its durability and longevity [26]. In another systematic review, bioactive modifications of dental implant surfaces such as calcium-phosphate coating seemed to be more beneficial for osseointegration and early peri-implant bone formation when compared to SLA or similar surfaces [27]. Many methods of nanometer alterations to implant surfaces have been documented such as TiO_2_ blasting followed by hydrofluoric acid treatment [28], Ion-beam-assisted deposition [29] and calcium phosphate nanoparticle modification by discrete CaP crystalline deposition [30]. By incorporating Ca ions to an SLA surface (XPEED^®^), an animal study showed significant improvement of the overall bone-to-implant contact and removal torque after 6 weeks, in comparison to control hydroxyapatite-blasted titanium implants [23]. The clinical efficiency of the nanostructured calcium-incorporated surface was also investigated by the authors in a clinical study attempting primary stability optimization by using fixtures with different thread depth depending on bone density [31]. Weekly implant stability quotients (ISQ) follow-up showed little to no drop in ISQ values at 3 weeks, especially for implants placed in low-density bone. The authors attributed this phenomenon to the biological effect of this surface in full expression leading to early bone deposition, particularly in cancellous bone where knife-edge threads cut into the trabeculae, causing minimal osteo-compression. Also, a study monitoring pre-osteoblastic cell behavior on machined (MAC) or grit-blasted Ti surfaces with and without Ca incorporation showed that Ca incorporation may have a favorable effect on the osseointegration of micro-structured Ti implants by promoting osteoblast proliferation and differentiation during the early healing phases after implantation [32]. Many clinical and radiographic evaluation methods have been developed for objective assessment of osseointegration (resonance frequency analysis, reverse torque analysis) [33,34,35]. Also, new experimental methods of texture and fractal dimension analyses seem promising in the evaluation of dental implants with complex geometry, but require further studies to further assess their reliability [36]. Therefore, one of the most objective techniques to evaluate implant surface performance is still the histomorphometric measurement and calculation of bone-implant contact (BIC) [37,38,39,40]. This technique has been successfully used in many studies to monitor, visualize, and assess bone apposition on the implant surface [41,42]. The main objective of the present study was to compare, at 4 and 6 weeks, BIC in implants having either machined surface, SLA medium roughness surface or a nanostructured calcium-incorporated surface (XPEED^®^)_._ The null hypothesis (*H0*) under test considered no statistically significant differences in BIC rate for SLA, MAC and XPEED^®^ surfaces after 4 and 6 weeks of healing time.

## 2. Materials and Methods

### 2.1. Study Design

This was a split-mouth randomized case/double-control histological human study.

Eleven patients, 6 females and 5 males (mean age 49.7 ± 13 years) with bilateral edentulous posterior maxilla and requiring implant therapy for fixed prosthetic rehabilitation were eligible for entering this study, provided that they fulfilled the f criteria reported in Table 1:

All procedures were performed in accordance with the recommendations of the Declaration of Helsinki for investigations with human subjects, as revised in Fortaleza [43]. All patients were thoroughly informed about the procedures and signed an informed consent form. The study was approved by the Ethics Committee at the Saint Joseph University of Beirut, Lebanon (USJ-2018-56). Preoperative evaluation included clinical examination of the edentulous ridges and natural dentition, as well as a cone beam computed tomography (CBCT) of the relevant sector. Patients underwent a prosthodontic evaluation for treatment planning and all surgeries were performed by the same experienced surgeon (C.M.) at the Oral Surgery Department, Faculty of Dental Medicine, Saint Joseph University (Beirut, Lebanon) between February 2019 and July 2021.

### 2.2. Surgical Procedure

Patients were asked to rinse with chlorhexidine digluconate solution (0.2%) for 1 min approximately 10 min before surgery. Under local anesthesia, a crestal incision and full-thickness flap elevation were performed, followed by standard implant preparation and placement of 2 regular implants per sector (AnyRidge, MegaGen, Gyeongsan, Republic of Korea). Then, each sector underwent drilling procedures with a 2.5 mm diameter final drill for the placement of 2 additional 3.5 × 8.5 mm mini-implants placed either between the 2 regular implants and/or distally to the most distal implant.

The mini-implants with the same geometry (MegaGen, Gyeongsan, Republic of Korea) had either XPEED^®^ surface (test), SLA surface (positive control) and MAC surface (negative control). This was an RCT with three-armed trial (XPEED^®^ vs. SLA vs. MAC) in which an allocation of 1:0.8:0.4 was used for ethical and economic reasons. Four patients with eight sectors received 16 XPEED^®^ implants, four patients with seven sectors received 13 SLA implants and three patients with six sectors received six MAC implants. Each sector received two mini-implants randomly chosen among the three groups. For the contralateral sector, implant positions by surface were inverted in order to reduce bias due to implant positioning and corresponding bone density. Soft tissues were approximated and sutured over it for a submerged healing protocol. Periapical radiographs were then performed following paralleling long cone technique.

Patients were prescribed analgesics and antibiotic coverage (amoxicillin 2 g/daily or in case of allergy clindamycin 600 mg/daily) for 7 days, as well as oral rinses of 0.12% chlorhexidine gluconate for 15 days following implant placement.

Patients were first recalled at 4 weeks for second stage surgery on one sector (chosen randomly), and before transgingival healing abutments were placed on the standard implants, mini-implants were retrieved using a trephine drill of 5.5 mm internal diameter.

At 6 weeks, 2nd stage surgery was performed for contra lateral side and, mini-implants were retrieved. All standard implants were prosthetically restored 1 month following second-stage surgery. Patients were then enrolled in a maintenance program and recalled every 4 months for periodontal and oral hygiene follow-up.

### 2.3. Histological Processing

To maintain the correct osmolarity at 278 mOsm/L, the retrieved biopsies were carefully rinsed with a cold 5% glucose solution to remove blood residues. They were then placed in a container with a 10% phosphate buffered formalin solution at pH 7.2 and sealed. The specimens remained in the formalin solution for two weeks.

After the fixation process was completed, specimens were placed under constant agitation in ascending concentrations of ethanol solutions as follows: 70% ethanol for one week; 80% ethanol for one week; 90% ethanol for one week and 100% ethanol for one week. After dehydration, the specimens were pre-infiltrated for 21 days in a 50% resin/alcohol solution (LR White, London Resin Co., Ltd., Aldermaston, UK), with three changes, followed by a week of infiltration in a 100% resin with two changes.

After 8 h of heat curing at 62 °C, the specimens were re-included in a light-curing hard resin (Technovit 7200 VLC, Kulzer, Wehrheim, Germany). After polymerization, undecalcified cut sections of 50 μm were prepared and ground down to about 30 μm using the TT System (TMA2, Grottammare, Italy). The sections for bone–implant contact measurements were stained with azure II/methylene blue and fuchsine acid. The investigation was carried out by means of a bright field light microscope (BX 51, Olympus America, Inc., Melville, NY, USA) connected to a high-resolution digital camera (FinePix S2 Pro, Fuji Photo Film Co., Ltd., Minato-Ku, Japan).

A histometric software package with image capturing capabilities (Image-Pro Plus 6.0, Media Cybernetics Inc., Bethesda, MD, USA) was used. To ensure accuracy, the software was calibrated for each experimental image using the ‘Calibration Wizard’, a feature that reports the number of pixels between two selected points of a micrometer scale. The linear remapping of pixel numbers in microns was used to calibrate the distance. All measurements were performed by the same experienced operator (T.T.).

### 2.4. Statistical Analysis

The statistical analyses were performed using IBM SPSS Statistics (IBM Corp, Armonk, NY, USA). Mean and standard deviation (SD) of bone–implant contact rate for different implant surfaces (SLA, MAC and XPEED^®^) after 4 and 6 weeks of healing were obtained. For each variable, variance normality and equality were assessed. The differences in mean values among the groups were analyzed. A two-way ANOVA followed by a post hoc Holm–Sidak test were applied for multiple comparisons. The threshold value to detect statistically significant differences was set at *p* < 0.05.

## 3. Results

### 3.1. Histological

A central section of each of the samples was analyzed and measured. However, for a more detailed description, two specimens for each group were reported.

Samples from the same group (MAC, SLA or XPEED) at different times (4 and 6 weeks) were reported in a sequential comparison. Four-week MAC samples (Figure 1) showed bone growth mainly related to distance osteogenesis. Osteoid tissue was mainly present in contact with native bone and only in some small area in direct contact with the implant surface. Six-week MAC samples (Figure 2) showed the same histological findings of Figure 1 with a distance osteogenesis bone-growth modality associated with an evident increase in newly formed bone along the implant surface. Four-week SLA samples (Figure 3) showed bone growth with contact osteogenesis modality. The newly formed bone was mainly present in direct contact with the implant surface. In almost all the samples, many vessels were noted in the tissues facing the implant interthread areas. Six-week SLA samples (Figure 4) showed a histological image resembling that of Figure 3, with diffuse contact osteogenesis along the implant surface and an increased amount of newly formed bone. Four-week XPEED samples (Figure 5) showed an advanced degree of osseointegration, most of the implant surfaces were covered by a layer of newly formed bone. In many implant interthread spaces, thin bone trabeculae in formation were visible, starting from the newly formed bone in direct contact with the implant surface. In the marrow spaces, many vessels were also present. Six-week XPEED samples (Figure 6) presented a histological aspect resembling that of Figure 5, with diffuse contact osteogenesis along the implant surface and an increased amount of newly formed bone.

### 3.2. Histomorphometry

The data collected in this study were evaluated for normal distribution using both the normality test (*p* = 0.528) and the test of equal variances (*p* = 0.123), demonstrating a normal distribution of the values (Table 2).

The two-way ANOVA test (Table 2 and Table 3) showed a greater difference in the mean values among the investigated types of surfaces than would be expected by chance, after allowing for effects of differences in time; there was a statistically significant difference (*p* < 0.001) among the groups. The difference in the mean values among the different levels of time was greater than would be expected by chance after allowing for effects of differences in surfaces; the difference was statistically significant (*p* < 0.001). The effect of different types of surfaces does not depend on what level of time was present. There was not a statistically significant interaction between surfaces and time. (*p* = 0.362). For the implants with MAC surface, the pairwise multiple comparison procedures using the Holm–Sidak method (Table 4) showed a BIC rate (±SD) of 16.8 (±5.0) after 4 weeks of healing (Figure 1), while the BIC rate on the same group of implants increased to 29.0 (±3.1) in the 6-week specimens (Figure 2). The group of implants with SLA surface showed a BIC rate of 18.5 (±2.3) at 4 weeks (Figure 3) and an increase to 33.7 (±3.3) at 6 weeks (Figure 4). The test group with XPEED^®^ surface presented rates of 22.4 (±1.3) and 38.6 (±3.2) at weeks 4 (Figure 5) and 6 (Figure 6), respectively. For all types of investigated surfaces, the time factor appeared to increase BIC rate from 19.2% at 4 weeks to 33.7% at 6 weeks. Also, surface type seemed to influence BIC rate with 22.9% for the MAC surface, 26.1% for SLA and 30.5% for XPEED^®^.

## 4. Discussion

Surface microtopography plays a major role in the speed and quality of osseointegration, therefore influencing the clinical behavior of dental implants [7]. This will mainly affect loading protocols and will also alter implant success rate, especially in soft bone [30].

The focus of the current study was to evaluate a nanostructured calcium-incorporated implant surface and histologically compare its performance to that of SLA and MAC surface implants with an identical implant geometry, placed in human subjects, and retrieved at either 4 or 6 weeks. All retrieved implants were placed in the posterior edentulous maxilla. Many studies showed the reduced bone quality in that region [44,45,46,47], that may result in reduced primary stability and delayed osseointegration [48]. It was shown that implant failure can be significantly increased when placed in the posterior soft bone, this may affect the results and therefore the loading protocols [49], making implant rehabilitation more delicate and less reliable. Comparing the BIC rate on implants with the same geometrical characteristics but with different surface topography, in the same bone quality, may provide a more objective assessment of the behavior of nanoscale surfaces under such a clinical condition.

While the first implants with machined surfaces were placed and loaded at 4- to 6-month intervals [1], more recent studies on modern implant surfaces showed very high success rates at a fraction of that time [50,51,52]. If the 6-week loading time point is considered the modern-day standard for implants with adequate surfaces [17,53], a 4-week loading protocol is still exclusive to implants presenting special surface topographies. However, it should be noted that studies assessing immediate loading of dental implants report high success rates and high predictability, but these are usually cases of edentulous patients requiring fixed full-arch prostheses where implants are splinted and thus present a favorable clinical situation [54]. As for data on immediately loaded implants in dentulous jaws, much less evidence was available, and there were no definite conclusions on the subject so far [55], even though a few recent studies showed promising results [56,57], and while soft tissue stability and marginal bone levels were comparable to those of early and conventional loading protocols, lower success rates were reported for immediately loaded implants [58]. Several studies showed the advantage of SLActive surface vs. SLA and the possibility of loading protocol decisions at either 4 or 6 weeks [50,59,60]. Other studies showed how surfaces affected ISQ values at different time points. Oates et al. [61], in a study monitoring early ISQ variations for SLActive (test) and SLA (control) surface implants, showed a shorter period of mean ISQ drop for the tested implants (lowest at 2 weeks) compared to the control (lowest at 4 weeks) (*p* < 0.0001), highlighting the enhanced osseointegration for the chemically modified surface.

In this study, implants were retrieved after 4 and 6 weeks; the rationale was to assess the different surface topography behavior at these two early healing phases, especially the implant osseointegration development in order to determine a safer and more objective time point for early loading protocol application.

In a previous study by Makary et al. [30], the ISQ of implants with Xpeed^®^ surface inserted in soft bone showed, at early healing phases, a virtual absence of ISQ decrease, which was attributed to the surface effect leading to early bone deposition.

Available long-term osseointegration studies in humans are commonly restricted to a small number of cases at a time and are mostly retrospective since they are usually performed on implants retrieved for reasons such as connection fracture [62,63] or at the end of an orthodontic treatment [64]. There is also a significant disparity of inclusion/exclusion criteria. Some authors resort to measures of corticalization or the inverse of bone index, which is a digital assessment of radiographic peri-implant bone texture that seems to be useful in evaluating its quality and evolution [65,66].

In a histologic and histomorphometric evaluation of two types of retrieved human titanium implants, a solid-screw titanium plasma-sprayed (TPS) implant removed 5 years after implantation due to implant/crown connection fracture and a sandblasted acid-etched titanium implant (SLA) removed after being used as anchorage for orthodontic treatment, the measured percentage of bone–implant contact around the SLA implant (76.6 ± 11.1) was significantly higher (*p* < 0.05) than that around the TPS implant (46.0 ± 5.5) [64].

In another publication, three functionally loaded, well-integrated nanotextured dental implants, were retrieved from the maxilla of a patient after 47 months of loading due to mechanical (not biological) failure. The results showed remodeled Haversian bone with BIC over 80% [67]. Also, in a case report on three dental implants with a sandblasted and acid-etched surface, retrieved after 45 months of function, BIC rates of 80.3% were reported [68]. Other authors conducted a study on 17 osseointegrated human dental implants with different surface types retrieved after a loading period of 4 to 20 years, found BIC values ranging from 32% to 83% [52]. Albrektsson et al. [69], in a histological analysis of more than 700 osseointegrated implants of various types, retrieved 6 months to 20 years after placement, observed BIC of >50% for maxillary implants compared to >75% for mandibular implants.

All the aforementioned publications have measured and compared BIC values for implants from 6 months to 20 years after placement. Additionally, implants were mostly retrieved for mechanical failure and many variables pertaining to geometry, surface and placement conditions/sites were not reported. It is therefore difficult to draw objective and relevant conclusions from these studies.

To the authors’ knowledge, few publications have described histomorphometric BIC results for retrieved implants in humans, in a split-mouth, posterior maxilla study design, comparing two or more implant surfaces, using fixtures with identical geometries retrieved at early healing phases.

A similar split-mouth human study on mini-implants retrieved at early healing stages was performed by Lang et al. [70]. However, the fixtures were placed in the mandibular third molar region, and the purpose was to compare regular SLA with modified SLA surfaces.

The results of the present investigation accept the null hypothesis under test for BIC rate comparison between SLA and MAC at 4 weeks, while rejecting the hypothesis under test in all other comparisons for surfaces and times, since they had a significant influence on the BIC rate.

MAC surfaces have shown limited performance at early healing phases and elevated failure rates in soft bone [71,72]. This surface is considered in the present work as a negative control. SLA surfaces have shown enhanced results over machined and seem to yield promising results, with higher success rates and faster loading time even in soft bone [18,19,73]. SLA was used as positive control. Recently, an animal study showed that incorporating Ca ions at a nanoscale on SLA surfaces (XPEED) resulted in higher BIC at early stages of healing [7,29].

When comparing BIC difference between weeks 4 and 6, all implant surfaces showed a significantly higher BIC at 6 weeks.

These results are coherent with the dynamics of bone healing around implant surfaces, since this phase is marked by woven bone formation and maturation, which is then gradually remodeled and replaced by lamellar bone [74]. Osseointegration is the consequence of a cascade of molecular and cellular events that occur after preparation of an implant bed and placement of a dental implant. It leads to the apposition of newly formed bone directly onto the implant surface. The pattern of bone formation and osseointegration seems to be similar to that observed in bone fracture healing [75]. Furthermore, the behavior of cells of osteogenic lineage is affected by microtopography and nanotopography of an implant surface [76].

In the aforementioned mandibular split-mouth study, implants were retrieved at either 7, 14, 28 or 42 days after implantation. When comparing their 4- and 6-week BIC values to our own, their numbers were higher [70]. This can be attributed to the fact that implants placed in mostly cortical mandibular bone will most probably have higher BIC rates than those inserted into posterior maxillary sites.

The comparison between the different types of surfaces after 4 weeks of healing showed a BIC average difference of 5.6% between XPEED^®^ and MAC. The result showed that the XPEED surface was significantly more osseointegrated than the MAC surface (*p* < 0.05). Analyzing the effect of Ca incorporation by the comparison of SLA vs. XPEED, there was also a significant increase in osseointegration of 3.9% (*p* < 0.05). The same comparison between the different surfaces at 6 weeks of healing showed that the average BIC difference between XPEED^®^ and MAC increased by 9.6% (*p* < 0.05). Also, after 6 weeks of healing, the XPEED^®^ group showed a significant increase in osseointegration of 4.9% (*p* < 0.05).

Similar conclusions were reported for nanostructured implant surfaces, with results showing that this kind of surface treatment enhances osteoblast adhesion, and therefore function, leading to better treatment outcomes [22].

However, when comparing the BIC rate of SLA versus MAC surfaces at 4 weeks, the difference in mean BIC between SLA and MAC (1.7%) appeared to be not statistically significantly different (*p* > 0.05) so, the null hypothesis for this comparison was confirmed. These findings may not be in line with most of the available literature comparing these two types of surface, since implants with SLA treatment became increasingly accepted as one of the most effective types of implant surface [52,77,78,79] and MAC surfaces are considered obsolete. In fact, it was shown that SLA results in a high degree of bone-to-implant contact and high removal torque values, as well as high osteoblasts differentiation, highlighting its osteoconductive properties, leading to enhanced bone formation and the possibility of reduced clinical healing times [19]. That being said, these values may still be relevant since the aforementioned BIC values are those of 4-week biopsies retrieved from human posterior maxillae. Most publications comparing BIC for these two surfaces are either animal studies [79,80], or follow different retrieval sites, such as the study by Lang et al. [70], where BIC values are higher than the present ones, but biopsies were all retrieved from posterior mandibles where bone density is significantly greater and is very likely to yield better results.

Overall, when considering the amount of osseointegration over time, BIC rate increase was on average (±SEM) 14.5% (0.7). When the results are split into 4- and 6-week categories and for each implant surface, MAC showed a mean increase (±SEM) of 12.2% (1.6), SLA 15.2% (1.1), and XPEED^®^ 16.2% (0.9). All differences were statistically significant. (*p* < 0.05). Otherwise, the interaction between surfaces and time appeared to be not statistically significant (*p* = 0.36). It is our opinion that this finding was mainly due to the short time interval considered in the present study and the subsequent intragroup variability of mineralization of the newly formed bone. This topic will be best addressed in a future paper on the same group of specimens which were treated with a bone labeling technique to determine the mineral apposition rate on the label (MARL).

From these results, it can be extrapolated that Ca incorporation may indeed enhance surface performance at early healing stages (both 4 and 6 weeks of healing) compared to SLA and MAC.

Our results were in line with those of Mangano et al. [47] who studied under a scanning electron microscope the bone/implant interface of an immediately loaded implant with XPEED^®^ surface, retrieved 1 month after function due to a traumatic injury. They reported that the implant was entirely surrounded by trabecular bone anchored to the metal surface.

Similar conclusions were reported by Suh et al. [20]. The authors, in an animal study where blasted Ti implant surfaces were compared to nanostructured, Ca-coated implant surfaces, reported a synergic effect on osseointegration due to their surface properties and biological activity.

Notably, these observations are corroborated with clinical results from our previous study [30].

This would also explain the non-significant difference in values for MAC and SLA, because a 4-week interval is apparently too short for surfaces without Ca incorporation to have elevated BIC, even for SLA.

Relative to the time factor (Figure 7), all implants showed a statistically significant difference between 4 and 6 weeks regardless of surface type.

After 6 weeks, comparisons showed that the time factor and surface type were always significant, particularly for XPEED^®^ surfaces which yielded 4-week BIC levels comparable to those of the MAC surface at 6 weeks. XPEED^®^ surfaces showed accelerated osseointegration by reaching high BIC levels within the first 4 weeks.

When compared to MAC implants, both XPEED^®^ and SLA surfaces seemed to achieve higher BIC at 4 and 6 weeks. However, XPEED^®^ surface also had a significantly higher rate of bone apposition compared to SLA at both intervals at 4 weeks (XPEED^®^: 22.4%–SLA: 18.5%) and 6 weeks (XPEED^®^: 38.6%–SLA: 33.7%). Notwithstanding the study design with split-mouth method and histological analysis confers great relevance and importance to our results, some study limitations emerge, mainly related to a monocentric study and the low number of samples which, perhaps, is plausible in research involving human subjects.

More studies are necessary in order to establish the timeline of the bone healing, early after implant placement.

## 5. Conclusions

Despite the scarcity of studies analyzing BIC values at such early stages, and overcoming the limitations of the present study, the following conclusions can be drawn:Nanostructured calcium-titanate-coated implant surfaces (XPEED^®^) showed higher BIC values at 4- and 6-week intervals.When compared to SLA and MAC surfaces, XPEED^®^ appeared to promote bone formation around the implant very early on after placement, even in soft trabecular bone of the posterior maxilla.Both SLA and XPEED^®^ surfaces showed bone formation with direct-contact osteogenesis.

## Figures and Tables

**Figure 1 nanomaterials-13-00357-f001:**
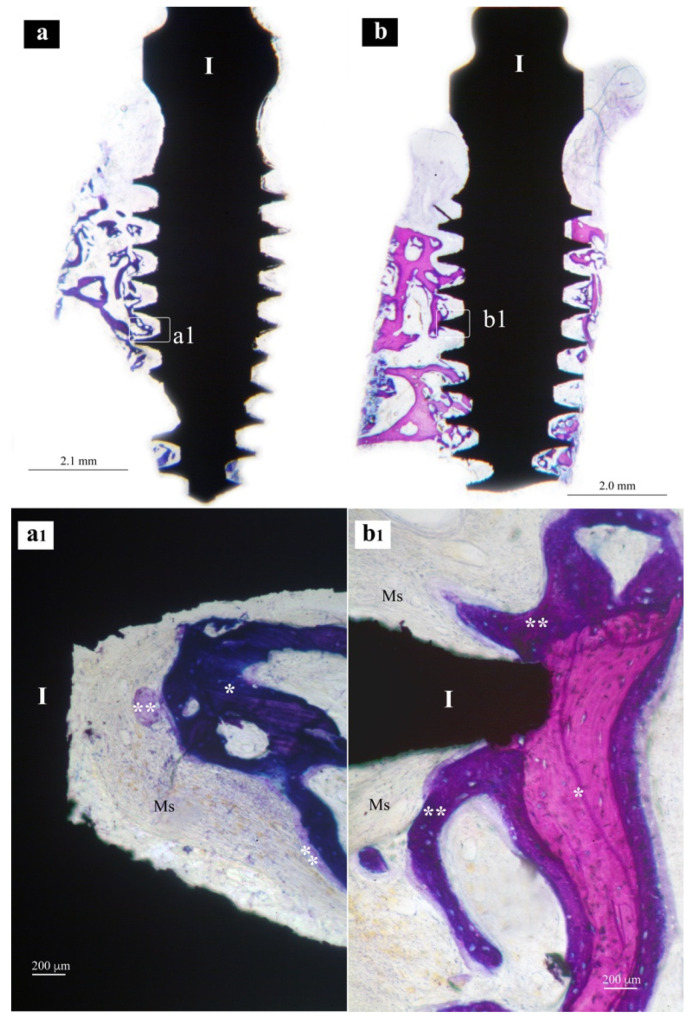
Central sections of two implants of the MAC group after 4 weeks of healing. In (**a**,**b**), original magnifications (×12), the implant bodies (I) appear to be partially surrounded by bone tissue in the threaded regions. In (**a1**,**b1**), original magnifications (×200), implant threads (I) appear to be in an early phase of osseointegration. Ms, marrow spaces; *, native bone; **, newly formed bone.

**Figure 2 nanomaterials-13-00357-f002:**
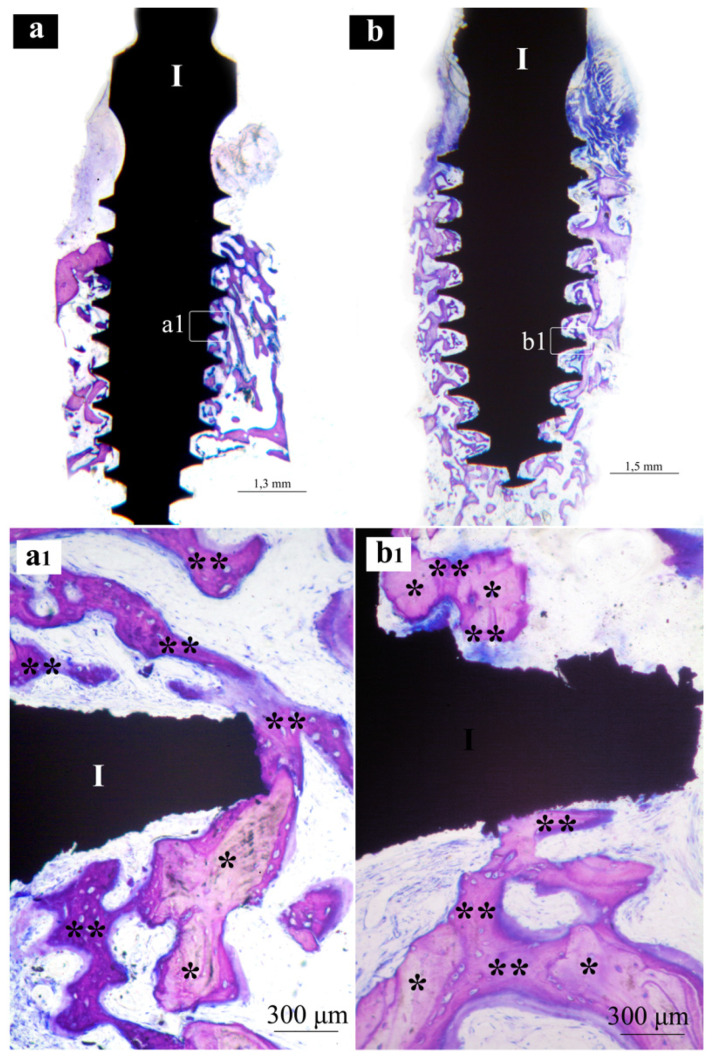
Central sections of two implants of the MAC group after 6 weeks of healing. In (**a**,**b**), original magnifications (×12), the implant bodies (I) appear to be, in the threaded regions, surrounded by much more bone tissue than in Figure 1. In (**a1**,**b1**), original magnifications (×200), the implant threads (I) appear to be in a more advanced phase of osseointegration. *, native bone; **, newly formed bone.

**Figure 3 nanomaterials-13-00357-f003:**
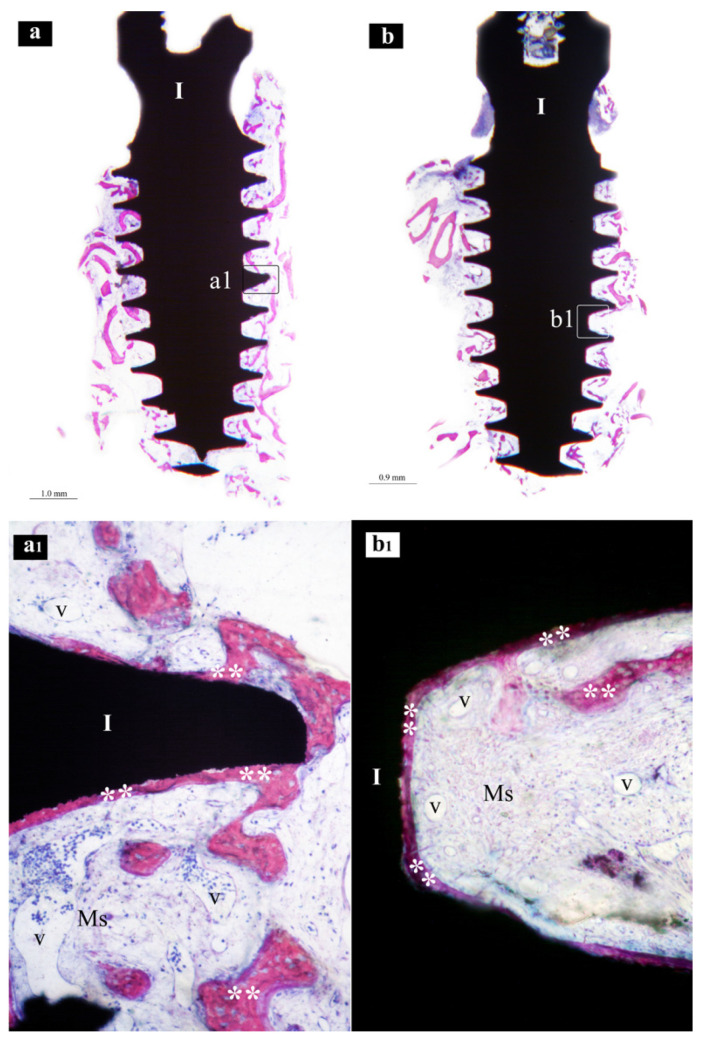
Central sections of two implants of the SLA group after 4 weeks of healing. In (**a**,**b**), original magnifications (×12), the implant bodies (I), mainly in the threaded regions, appear to be partially surrounded by mineralized bone. In (**a1**), original magnifications (×200), the implant threads (I) appear to be surrounded by newly formed bone (**), while in the marrow spaces (Ms) several vessels (v) in section are visible. In (**b1**), original magnifications (×200), the implants (I) in the inter-thread space newly formed bone (**) appears to be formed in contact with the implant surface, while in the marrow spaces (Ms) several vessels (v) are visible.

**Figure 4 nanomaterials-13-00357-f004:**
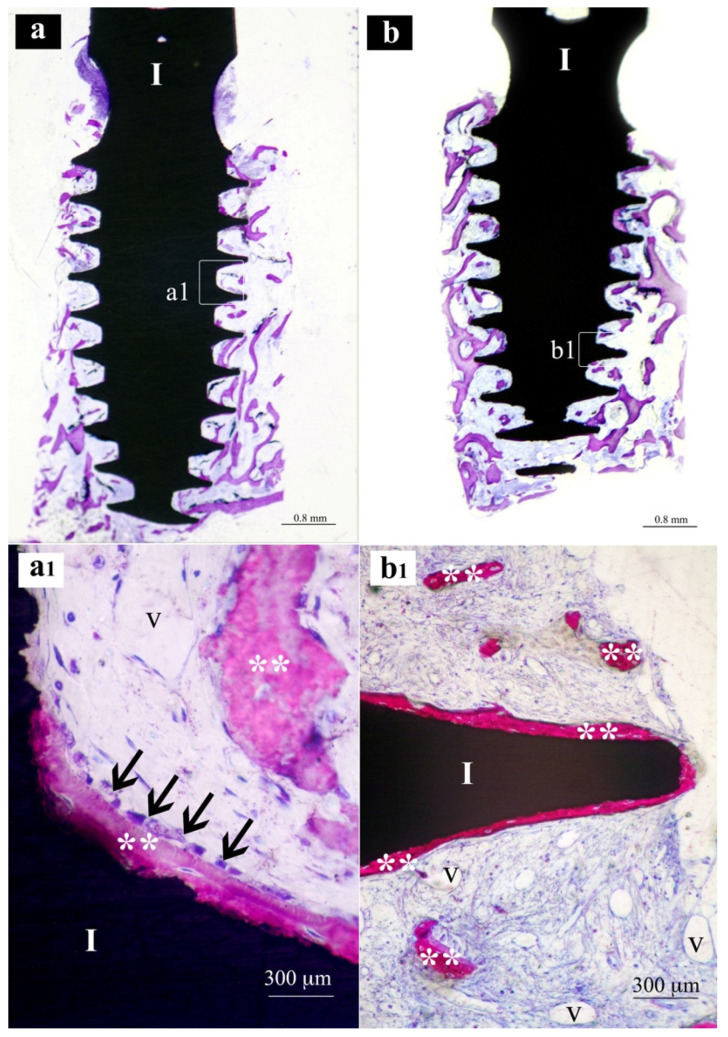
Central sections of two implants of the SLA group after 6 weeks of healing. In (**a**,**b**), original magnifications (×12), the implant bodies (I) appear to be, in the threaded regions, surrounded by much more bone tissue than in Figure 3. In (**a1**,**b1**), original magnifications (×200), the implant threads (I) appear to be in a more advanced phase of osseointegration. **, newly formed bone; V, vessel; black arrows, osteoblastic cells.

**Figure 5 nanomaterials-13-00357-f005:**
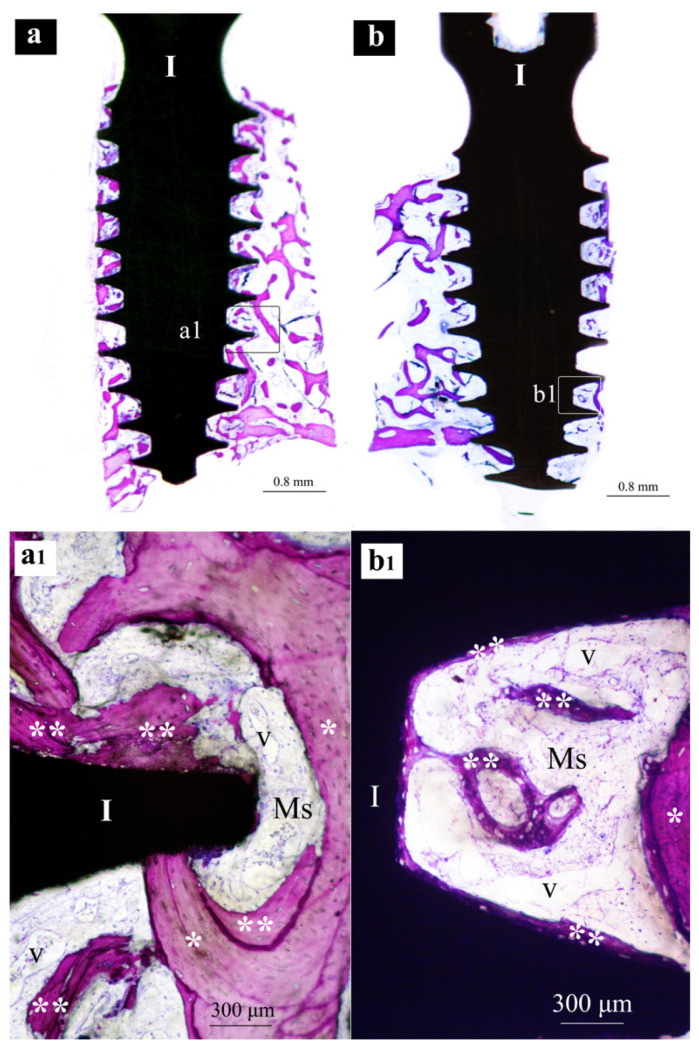
Central sections of two implants of the XPEED group after 4 weeks of healing. In (**a**,**b**), original magnifications (×12), the implant bodies (I), along the entire threaded regions, appear to be surrounded by mineralized bone. In (**a1**), original magnifications (×200), the implant threads (I) appear to be in contact with both newly formed bone (**) and native bone trabeculae (*), while in the marrow spaces (Ms) enlarged vessels (v) are present near the implant surface, In (**b1**), original magnifications (×200), the implants (I) in the inter-thread space, a thin layer of newly formed bone (**) appear to be in contact with the implant surface forming struts of bone toward area of native bone (*). In the marrow spaces (Ms) several enlarged vessels (v) are visible.

**Figure 6 nanomaterials-13-00357-f006:**
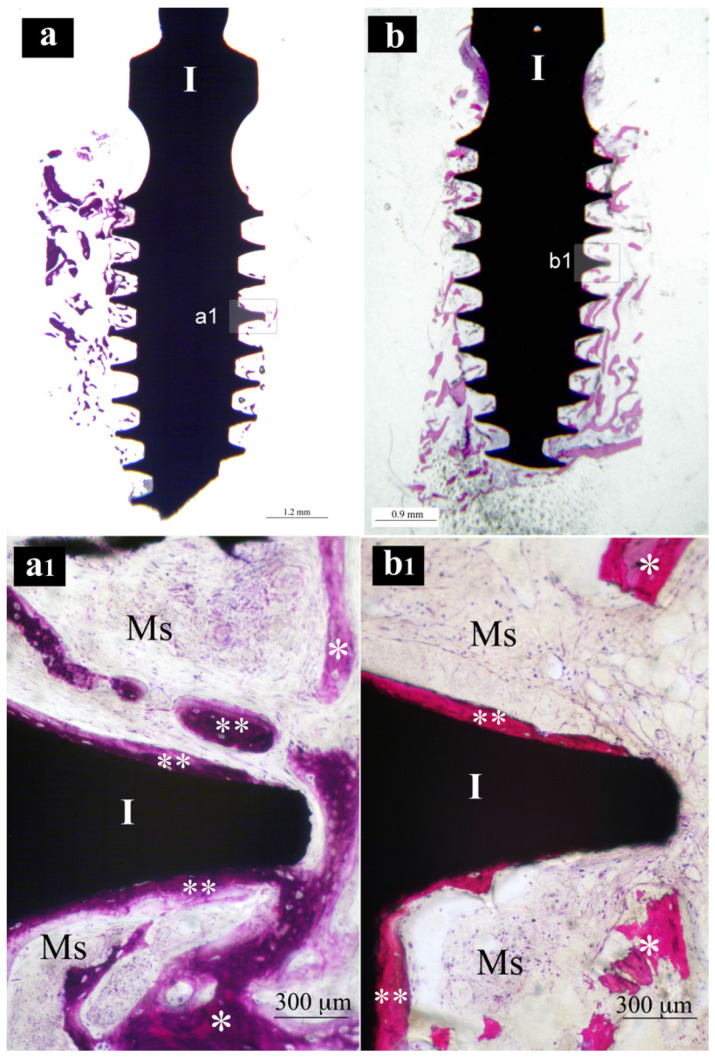
Central sections of two implants of the XPEED group after 6 weeks of healing. In (**a**,**b**), original magnifications (×12), the implant bodies (I), along the entire threaded regions, appear to be surrounded by an increase amount of mineralized bone compared to Figure 5. In (**a1**), original magnifications (×200), the implant threads (I) appear to be in contact with a layer of newly formed bone (**); Ms, marrow space; *, native bone. In (**b1**), original magnifications (×2000), the implant threads (I) appear to be almost completely harvested by thin layer of newly formed bone (**) with islands of native bone (*).

**Figure 7 nanomaterials-13-00357-f007:**
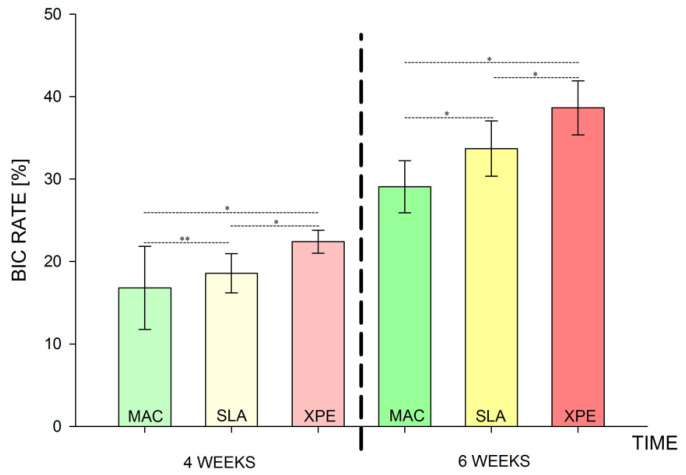
Graph reporting the BIC rate weighted for time. * statistically significant (*p* < 0.05); ** not statistically significant (*p* > 0.05).

**Table 1 nanomaterials-13-00357-t001:** Inclusion and exclusion criteria adopted.

Inclusion Criteria	Exclusion Criteria
Height of the residual bone crest in the programmed implant site ≥ 9 mm	Myocardial infarction within the past 6 months.
Thickness of the residual bone crest in the programmed implant site ≥ 7 mm.	Poorly controlled diabetes (HBA1c > 7.5%).
Availability, in each sector, of sufficient mesio-distal space allowing placement of 2 standard-sized implants and at least 2 mini-implants (3.5 × 8.5 mm) for retrieval.	Coagulation disorders.
Healed bone crest (≥3 months elapsed after extraction or tooth loss).	Radiotherapy to the head/neck area within the past two years.
Age > 18 years.	Present or past treatment with intravenous bisphosphonates.
	Immunocompromised patients.
Ability to examine and fully understand the study protocol.	Psychological or psychiatric problems.
	Alcohol or drug abuse.
	Poor oral hygiene and motivation (full mouth plaque score > 30% and/or full mouth bleeding score > 20%).
	Uncontrolled periodontal disease.

**Table 2 nanomaterials-13-00357-t002:** Summary of the statistical results for two-way ANOVA.

*Normality Test: Passed (p = 0.528)*					
*Equal Variance Test: Passed (p = 0.123)*					
*Source of Variation*	DF *	SS *	MS *	F *	*p*
*SURFACES*	2	290,982	145,491	17,956	<0.001
*TIME*	1	1,528,390	1,528,390	188,626	<0.001
*SURFACES × TIME*	2	17,050	8525	1052	0.362
*Residual*	29	234,980	8103		
*Total*	34	2,481,227	72,977		
*Power of performed test with alpha = 0.0500: for SURFACES: 1000*					
*Power of performed test with alpha = 0.0500: for TIME: 1000*					

* DF degree of freedom for each origin of variance; SS is the sum of squares for each origin of variance; MS is the mean square; F is the mean of intragroup variance.

**Table 3 nanomaterials-13-00357-t003:** Summary of least square means of BIC for groups that are adjusted for means of surfaces, time, surface x time.

*Least Square Means for SURFACES **^1^*		
*Groups* ***	Mean	SEM
*SLA (n = 13)*	26,131	0.811
*XPEED^®^ (n = 16)*	30,514	0.717
*MACHINED (n = 6)*	22,933	1162
** *Least square means for TIME **^2^* **		
*Groups **	Mean	SEM
*4W (n = 18)*	19,254	0.716
*6W (n = 17)*	33,798	0.780
** *Least square means for SURFACES × TIME **^3^* **		
*Groups **	Mean	SEM
*SLA × 4W (n = 7)*	18,562	1006
*SLA × 6W (n = 6)*	33,700	1273
*XPEED^®^ × 4W (n = 8)*	22,400	0.949
*XPEED^®^ × 6W (n = 8)*	38,629	1076
*MAC × 4W (n = 3)*	16,800	1643
*MAC × 6W (n = 3)*	29,067	1643

* BIC rate. ** Grouping factor/s used to compute the comparison. ^1^ grouped for surfaces, out of time interval. ^2^ grouped for time interval, out of different surfaces. ^3^ grouped for surface and time.

**Table 4 nanomaterials-13-00357-t004:** Summary of all Pairwise Multiple Comparison Procedures (Holm–Sidak method): Overall significance level = 0.05.

*Comparison*	Diff of Means	t	Unadjusted P	Critical Level	Significant?
*Comparisons for factor: **SURFACES***					
*XPEED^®^* vs. *MAC*	7581	5551	0.00000549	0.017	Yes
*XPEED^®^* vs. *SLA*	4383	4047	0.000352	0.025	Yes
*SLA* vs. *MAC*	3198	2256	0.0318	0.050	Yes
*Comparisons for factor: **TIME***					
*6W* vs. *4W*	14,544	13,734	3191 × 10^−14^	0.050	Yes
*Comparisons for factor: **TIME within SLA***					
*6W* vs. *4W*	15,138	9328	0.000	0.050	Yes
*Comparisons for factor: **TIME within XPEED^®^***					
*6W* vs. *4W*	16,229	11,313	0.000	0.050	Yes
*Comparisons for factor: **TIME within MAC***					
*6W* vs. *4W*	12,267	5278	0.000	0.050	Yes
*Comparisons for factor: **SURFACES within 4W***					
*XPEED^®^* vs. *MAC*	5600	2951	0.006	0.017	Yes
*XPEED^®^* vs. *SLA*	3837	2774	0.010	0.025	Yes
*SLA* vs. *MAC*	1763	0.915	0.368	0.050	No
*Comparisons for factor: **SURFACES within 6W***					
*XPEED^®^* vs. *MAC*	9562	4868	0.000	0.017	Yes
*XPEED^®^* vs. *SLA*	4929	2957	0.006	0.025	Yes
*SLA* vs. *MAC*	4633	2229	0.034	0.050	Yes

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
