# Peer review of "Nanostructured Calcium-Incorporated Surface Compared to Machined and SLA Dental Implants—A Split-Mouth Randomized Case/Double-Control Histological Human Study"

_nanomaterials, 2023, doi:10.3390/nano13020357_

Round 1

Reviewer 1 Report

The article is written in a careful, clear and effective way. Nevertheless, tracking the research results and following the thread can be difficult with too many small histological figures that should be combined in 2, maximum 3.

Same with tables. It is much easier to view the results of statistical analyzes presented in the form of graphs, so I suggest changing these data to graphs.

Author Response

REVIEWER 1

R: thank you so much for the evaluation

Comments and Suggestions for Authors

The article is written in a careful, clear and effective way. Nevertheless, tracking the research results and following the thread can be difficult with too many small histological figures that should be combined in 2, maximum 3. 

R: the dimension and resolution of the Histo figures is very High 4k x 5k pixel (34 x 51cm for each set of images) at 300 pixel/inch, so the editorial staff will be able to choose the more appropriate dimension. In the manuscript's revision version, obviously, the limited dimension of the figures is due to the high resolution of the images that are reduced to a readable dimension.

Same with tables. It is much easier to view the results of statistical analyzes presented in the form of graphs, so I suggest changing these data to graphs.

R: the data were already reported in a graph (fig. 7)  presented in the discussion section to improve the manuscript readability.

Reviewer 2 Report

Dear Authors,

thank you for the well prepared manuscript. I would add some of the "improvements" though:

1. Please, search the "Coatings" journal for the latest papers connected to that topic to incorporate them in the introduction 

2. In the introduction, please add the information on the connection between bacterial contamination and the coatings, see:

Kubasiewicz-Ross P, Fleischer M, Pitułaj A, Hadzik J, Nawrot-Hadzik I, Bortkiewicz O, Dominiak M, Jurczyszyn K. Evaluation of the three methods of bacterial decontamination on implants with three different surfaces. Adv Clin Exp Med. 2020 Feb;29(2):177-182. doi: 10.17219/acem/112606.

also when the fractal and texture analyses are taken into account, eg. 

Hadzik J, Kubasiewicz-Ross P, Simka W, Gębarowski T, Barg E, Cieśla-Niechwiadowicz A, Trzcionka Szajna A, Szajna E, Gedrange T, Kozakiewicz M, Dominiak M, Jurczyszyn K. Fractal Dimension and Texture Analysis in the Assessment of Experimental Laser-Induced Periodic Surface Structures (LIPSS) Dental Implant Surface-In Vitro Study Preliminary Report. Materials (Basel). 2022 Apr 7;15(8):2713. doi: 10.3390/ma15082713.

This is a novel but easy method of establishing on the implant stability and to my mind would be worth mentioning. 

3. Please add inclusion and exclusion criteria in form of the table or a graph

4. In supplementary materials / informations, the approvement of the Ethical Comitee (with a number) should be mentioned. Please, leave it in the materials and methodology as well.

5. In the discussion section, bone index should be mentioned as the method of possible diagnostics. It should also refer to the mentioned surface and fractal analyses

6. In the discussion section, please add more advanced information on the survival rate of implants, when the implant is loaded immediately, eg.:

Silva AS, Martins D, Sá J, Mendes JM. Clinical evaluation of the implant survival rate in patients subjected to immediate implant loading protocols. Dent Med Probl. 2021;58(1):61–68. doi:10.17219/dmp/130088

Krawiec M, Olchowy C, Kubasiewicz-Ross P, Hadzik J, Dominiak M. Role of implant loading time in the prevention of marginal bone loss after implant-supported restorations: A targeted review. Dent Med Probl. 2022;59(3):475–481. doi:10.17219/dmp/150111

7. Beside the mentioned aspects that should be incorporated to the introduction, I would also add the influence of natural polymers for implant healing, eg.:

Paradowska-Stolarz A, Wieckiewicz M, Owczarek A, Wezgowiec J. Natural Polymers for the Maintenance of Oral Health: Review of Recent Advances and Perspectives. Int J Mol Sci. 2021 Sep 25;22(19):10337. doi: 10.3390/ijms221910337.

8. The figure 7 should be in the results section

9. Please, add the limitations of the study.

After that changes, the paper should be reevaluated.

Author Response

REVIWER 2

R: thank you very much for the evaluation

Comments and Suggestions for Authors

Dear Authors,

thank you for the well prepared manuscript. I would add some of the "improvements" though:

  1. Please, search the "Coatings" journal for the latest papers connected to that topic to incorporate them in the introduction 

R: the introduction section was improved reporting the results of the following studies:

Paradowska-Stolarz A, Wieckiewicz M, Owczarek A, Wezgowiec J. Natural Polymers for the Maintenance of Oral Health: Review of Recent Advances and Perspectives. International journal of molecular sciences. 2021 Sep 25;22(19):10337.)

Kubasiewicz-Ross P, Fleischer M, Pitułaj A, Hadzik J, Nawrot-Hadzik I, Bortkiewicz O, Dominiak M, Jurczyszyn K. Evaluation of the three methods of bacterial decontamination on implants with three different surfaces. Advances in Clinical and Experimental Medicine. 2020;29(2):177-82).

Vishwakarma V, Kaliaraj GS, Amirtharaj Mosas KK. Multifunctional Coatings on Implant Materials—A Systematic Review of the Current Scenario. Coatings. 2023 Jan;13(1):69).

Vishwakarma V, Kaliaraj GS, Amirtharaj Mosas KK. Multifunctional Coatings on Implant Materials—A Systematic Review of the Current Scenario. Coatings. 2023 Jan;13(1):69).

(Hadzik J, Kubasiewicz-Ross P, Simka W, GÄ™barowski T, Barg E, CieÅ›la-Niechwiadowicz A, Trzcionka Szajna A, Szajna E, Gedrange T, Kozakiewicz M, Dominiak M. Fractal Dimension and Texture Analysis in the Assessment of Experimental Laser-Induced Periodic Surface Structures (LIPSS) Dental Implant Surface—In Vitro Study Preliminary Report. Materials. 2022

  1. In the introduction, please add the information on the connection between bacterial contamination and the coatings, see:

- Kubasiewicz-Ross P, Fleischer M, PituÅ‚aj A, Hadzik J, Nawrot-Hadzik I, Bortkiewicz O, Dominiak M, Jurczyszyn K. Evaluation of the three methods of bacterial decontamination on implants with three different surfaces. Adv Clin Exp Med. 2020 Feb;29(2):177-182. doi: 10.17219/acem/112606.

R: done

also when the fractal and texture analyses are taken into account, eg. 

- Hadzik J, Kubasiewicz-Ross P, Simka W, GÄ™barowski T, Barg E, CieÅ›la-Niechwiadowicz A, Trzcionka Szajna A, Szajna E, Gedrange T, Kozakiewicz M, Dominiak M, Jurczyszyn K. Fractal Dimension and Texture Analysis in the Assessment of Experimental Laser-Induced Periodic Surface Structures (LIPSS) Dental Implant Surface-In Vitro Study Preliminary Report. Materials (Basel). 2022 Apr 7;15(8):2713. doi: 10.3390/ma15082713.

R: done

This is a novel but easy method of establishing on the implant stability and to my mind would be worth mentioning. 

  1. Please add inclusion and exclusion criteria in form of the table or a graph

R: the table was created and inserted as requested

  1. In supplementary materials / informations, the approvement of the Ethical Comitee (with a number) should be mentioned. Please, leave it in the materials and methodology as well.

R: the requested revision was made and statements of the approvement of the Ethical Committee with a number was added

  1. In the discussion section, bone index should be mentioned as the method of possible diagnostics. It should also refer to the mentioned surface and fractal analyses

R: done

  1. In the discussion section, please add more advanced information on the survival rate of implants, when the implant is loaded immediately, eg.:

- Silva AS, Martins D, Sá J, Mendes JM. Clinical evaluation of the implant survival rate in patients subjected to immediate implant loading protocols. Dent Med Probl. 2021;58(1):61–68. doi:10.17219/dmp/130088

- Krawiec M, Olchowy C, Kubasiewicz-Ross P, Hadzik J, Dominiak M. Role of implant loading time in the prevention of marginal bone loss after implant-supported restorations: A targeted review. Dent Med Probl. 2022;59(3):475–481. doi:10.17219/dmp/150111

R: the studies were reported and discussed

  1. Beside the mentioned aspects that should be incorporated to the introduction, I would also add the influence of natural polymers for implant healing, eg.:

- Paradowska-Stolarz A, Wieckiewicz M, Owczarek A, Wezgowiec J. Natural Polymers for the Maintenance of Oral Health: Review of Recent Advances and Perspectives. Int J Mol Sci. 2021 Sep 25;22(19):10337. doi: 10.3390/ijms221910337.

R: this aspect was considered

  1. The figure 7 should be in the results section

R: it is opinion of the authors that, due to the presence in the result section of tables, it is not useful to move the graph in that section (redundant data) instead we believe that in the discussion section a graph, that summarize the statistical results, can improve the readability of the study and gives to the readers a clear vision on the conclusions.

  1. Please, add the limitations of the study.

R: a sentence at the end of the discussion section was added to declare the limitation of the study.

After that changes, the paper should be reevaluated.

Reviewer 3 Report

This paper reports the results of placing dental implants with three different surfaces in humans and monitoring the initial healing process. Before publication, several corrections are needed.

Line 74: typo (scid -> acid)

Line 90: The full name of the ISQ must be written.

Fig 1~ 6: Mark which part of the image above is the enlarged a1, b1 image.

Fig. 6b is not visible as the central section. Is there a reason why the central section image was not included?

Author Response

REVIEWER 3

R: thank you very much for the evaluation

Comments and Suggestions for Authors

This paper reports the results of placing dental implants with three different surfaces in humans and monitoring the initial healing process. Before publication, several corrections are needed.

Line 74: typo (scid -> acid)

R: correction was made

Line 90: The full name of the ISQ must be written.

R: correction was made, ISQ was spelled out

Fig 1~ 6: Mark which part of the image above is the enlarged a1, b1 image.

R: all figures were improved following the suggestions

Fig. 6b is not visible as the central section. Is there a reason why the central section image was not included?

R: the image was changed with a central section (the image is not of good quality)

Round 2

Reviewer 2 Report

Thank you for the corrections and explanations. Good luck with publishing the article

Author Response

REVIEWER 2

Comments and Suggestions for Authors

Thank you for the corrections and explanations. Good luck with publishing the article

R: thank you so much for your constructive revision